# CRYPTOGRAPHY IN SEMANTIC WATERMARKS: UNDETECTABILITY AND DEPLOYMENT IMPLICATIONS

## ABSTRACT

Semantic watermarking methods enable the direct integration of watermarks into the generation process of latent diffusion models by only modifying the initial latent noise. One group of watermarks such as Gaussian Shading (GS) and Pseudo-random Codes Watermarks (PRCW) relies on cryptographic primitives to ensure provable undetectability. However, we find that the use of randomness in these schemes has pitfalls, which leads to a flaw in the proof of Gaussian Shading and to ambiguity in the literature. We propose a novel, general framework based on IND\$-CPA security which highlights the effect of randomness and reveals that reusing it makes watermarks trivially detectable. As a direct consequence, we obtain an undetectable but inefficient deployment mode for GS. To regain practicability, we propose several speed-ups for GS and provide extensive experiments to compare those with other undetectable watermarks in robustness, speed, and quality.

## 1 INTRODUCTION

The rapid progress of generative AI for image generation—particularly latent diffusion models (LDMs)—has made it increasingly difficult to distinguish real and synthetic media. While these technologies enable valuable applications, they also create new risks. Deepfakes, for instance, are now being used to commit fraud, spread disinformation, and manipulate public opinion (Marchal et al., 2024). This creates an urgent need to identify whether media is synthetic and to trace which system or user produced it. Watermarking has emerged as a promising strategy for this purpose, and major players such as Google are already deploying it (Google DeepMind, 2024).

Semantic watermarks based on diffusion inversion are a novel watermark paradigm for LDM-generated images (Wen et al., 2023; Yang et al., 2024b; Ci et al., 2024; Gunn et al., 2025). Here, a specific watermark pattern is embedded into the initial latent noise during generation, so that the watermark becomes deeply rooted in the generated image and is expressed via high-level features such as image composition. This is in contrast to common post-hoc watermarks which are added as an imperceptible noise pattern on top of images. To check for watermark presence, the denoising process in a diffusion model is inverted and the recovered latent noise is checked for the watermark pattern. Importantly, these methods do not require training and support plug-and-play deployment.

The semantic watermark methods differ in how they modify the initial latent noise ($z_T$) and can be categorised into two groups. *Distribution-changing* methods such as Tree-Ring (Wen et al., 2023) and RingID (Ci et al., 2024) add fixed circular patterns into the frequency spectrum of $z_T$. This changes the distribution of $z_T$, and consequently, the distribution of the generated images. In contrast, *distribution-preserving* methods such as Gaussian Shading (Yang et al., 2024b, GS) and Pseudo Random Code Watermarking  (Gunn et al., 2025, PRCW) keep the distribution of $z_T$ unchanged. For this, they critically rely on *cryptographic primitives* to generate a pseudorandom sequence that steers the sampling process of $z_T$.

However, we find that the cryptographic foundation is not properly specified and that practical deployment considerations have been overlooked in previous work. In this work, we therefore clarify key aspects of the use of cryptographic primitives for semantic watermarking and urge to include practical deployment considerations. More specifically, we found some problems with the proof provided by Yang et al. (2024b), as well as an inexact cryptographic specification, leading to open questions about practical deployment. To address these shortcomings, we provide a novel proof of

undetectability (i.e. that watermarked images are indistinguishable from images without watermark) and propose modifications that allow for correct usage in practice. In detail, our contributions are as follows:

- Theoretical insights for cryptographic schemes: We revisit the usage of cryptography in semantic watermarking providing a proper foundation for future research. We provide a general framework for their construction and give a new proof of undetectability based on IND$-CPA which requires the correct use of randomness.

- Deployment: We outline the implications on practical deployment resulting from the proposed cryptographic framework. Our analysis shows that the randomness requirements are not fulfilled in the current deployment of GS. Therefore, we propose techniques to fix this issue while still allowing for an efficient runtime.

- Evaluation: We perform a comprehensive evaluation comparing undetectable watermarks in terms of runtime, watermark verification, robustness against common image transformations, and impact on generation quality.

## 2 BACKGROUND

We shortly revisit the relevant basics of LDMs, coding theory, and cryptography. Then, we build a framework to describe recently proposed semantic watermarking methods that rely on cryptography and that thus fall into the scope of our work.

### 2.1 LATENT DIFFUSION MODELS AND SAMPLING

Latent Diffusion Models (LDM) are a class of generative models that operate in the latent space of an image autoencoder. A diffusion model generates data by reversing a gradual noising process. During generation, first an initial latent $z_T \sim \mathcal{N}(0, I)$ is sampled. Subsequently, multiple iterations of denoising are applied. In every step, the image $z_t$ with a noise strength associated with denoising step $t$ is used to predict a slightly less noisy version $z_{t-1}$. The final latent $z_0$ is then passed through the VAE decoder to produce the final pixel image $x$. The image generation process can be conditioned, e. g., on a prompt $\pi$. We denote the entire generation process by $\mathcal{G}(z_T, \pi) = x$. Various samplers have been proposed, such as the prominent deterministic DDIM sampler (Ho et al., 2020).

DDIM Inversion (Mokady et al., 2023) tries to invert the generation process and to recover a matching latent $\hat{z}_T$ to a given image. At each inversion step, it tries to find the most likely previous latent $\hat{z}_t$ given $\hat{z}_{t-1}$. We denote this inversion process as $\mathcal{I}(x) = \hat{z}_T$. The inversion does not consider the prompt and performs some approximations so that the original latent is not necessarily recovered.

However, exact Inversion (Hong et al., 2024) improves inversion accuracy at the cost of much slower runtime due to optimisation within every inversion step. reduces the added reconstruction noise. While it still does not take the prompt into account, at each step it runs gradient descent to account for some of the dropped factors. While more exact, this procedure is a lot slower than DDIM Inversion.

### 2.2 CODING THEORY AND CRYPTOGRAPHY

**Pseudorandomness** A bit string is called *pseudorandom* if one cannot distinguish it from a truly random bit string in polynomial time with non-negligible success probability. An algorithm that is used to generate pseudorandom numbers is called a pseudorandom number generator (PRNG).

**Stream Cipher** A stream cipher is an encryption algorithm that encrypts a message $m$ one bit at a time, thus allowing for immediate encryption and decryption and arbitrary message sizes. During **encryption**, a stream cipher encrypts the message $m$ using bitwise XOR. That is, the encrypted message (i.e. the ciphertext) is given by $c = K \oplus m$. Here, $K = \mathrm{PRNG}(k, \eta)$ is a sequence of pseudorandom bits referred to as the keystream, where $k$ is the secret key and $\eta$ is a public nonce. Because $K$ is a pseudorandom bit string, $c$ must also be pseudorandom. The pseudorandomness of $c$ can be used to achieve practically distribution-preserving sampling while carrying a message (like in Gaussian Shading Yang et al. (2024b)). For **decryption**, the receiver can recover the original

message using bitwise XOR: $m = K \oplus c$. To do so, the receiver must generate the exact same keystream $K$ and thus needs to know the secret key $k$ and the nonce $\eta$.

**Error Correcting Codes**  An error correcting code (ECC) is a method that allows the encoded message $c$ to be slightly altered but still be decoded to the original message $m$. A simple ECC consists of replicating the message $\rho$ times, $\text{Repl}_\rho(m) = m||m||\ldots||m$, which can be decoded by majority voting at every position.

State-of-the-art ECCs with real-world applications (WiFi, 5G) are among others low-density parity-check (LDPC) codes (Gallager, 1962). An LDPC is defined by the parity-check matrix $H \in \{0,1\}^{(|c|-|m|)\times|c|}$ that is chosen to be sparse for efficient decoding. The encoded message $c$ consists of the original message $m$ concatenated with parity bits. The parity-check matrix $H$ is used to enforce that the encoded messages $c \in \{0,1\}^{|c|}$ satisfy $Hc^T = 0 \pmod 2$. For encoding, a generator matrix $G$ can be derived from $H$ and the encoded message is computed by $c = mG \pmod 2$. $G$ is defined such that $c$ satisfies the constraints defined by $H$. Decoding uses belief propagation in order to reconstruct the uncorrupted $c$ by modifying the received $c$ to fit this constraint.

## 2.3 INVERSION-BASED SEMANTIC WATERMARKING

Watermarks typically operate in a post-hoc manner by adding an imperceptible pattern on top of an image. The generation of images from scratch, however, provides the possibility to create *in-generation* watermarks that are already inserted during the generation process. Recently proposed semantic watermarking methods alter the LDM generation process by embedding a watermark pattern into the initial latent $z_T$ (Wen et al., 2023; Yang et al., 2024b; Gunn et al., 2025; Yang et al., 2025). Thus, after generation, the watermark becomes realised as higher-level image features, such as the image's composition, object shapes, and edge details. The watermark's presence is verified by inverting the denoising process and by checking for the watermark in the recovered latent noise.

Formally, we define a **watermarked generation algorithm** as a function $\text{WM}_{k,\mathcal{M}}(m,\eta,\pi) = x$. The inputs are a message $m$, a nonce $\eta$ potentially, and a prompt $\pi$. The output is a watermarked image $x$. Internally, the function WM uses a secret key $k$ and a generative model $\mathcal{M}$ (Zhao et al., 2025). For this work, we consider the verification of a watermark as the extraction of its message, i.e., $\hat{m} = \text{WM}_{k,\mathcal{M}}^{-1}(x)$.

**Distribution-Preserving Semantic Watermarking**  In this work, we focus on the group of distribution-preserving semantic watermarks. They rely on a cryptographic algorithm to drive the sampling of a watermarked $z_T$ that is then used for generation. The cryptographic algorithms allow generating pseudorandom bitstrings, which in turn can be exploited to sample $z_T$ from the original unwatermarked initial latent distribution (typically $\mathcal{N}(0,\text{I})$). This ensures that the generator's output distribution is practically identical to the unwatermarked distribution[1]. Informally, a watermark is called *undetectable* if it is impossible to distinguish between multiple watermarked and unwatermarked images in polynomial time without access to the secret. This also implies that there is no quality metric (on polynomially many samples) that measures any difference. A watermark is called *distortion free* if it is impossible to distinguish it based on a single image.

Watermarking schemes based on cryptographic primitives generally consists of three modules as depicted in Figure 1 (top part). The first is a cryptographic unit that takes as input a message $m$, a secret key $k$ and (potentially) a nonce $\eta$, and outputs a long bit string $c$ (e.g. as many bits as latent pixels). The output fulfils the following properties: It should be *pseudorandom*, such that it hides the original message cryptographically, and it should be *robust* to perturbations, meaning that one can recover the original message even in the presence of a certain amount of bit flips. The second module is a sampler $\mathcal{S}$. It takes the bit string from the cryptographic unit and uses it to sample an initial latent $z_T$ that follows a standard Gaussian distribution. It is crucial that the sampler produces an unbiased Gaussian output given an unbiased binary input. The final module is the normal generation procedure $\mathcal{G}$ of the model $\mathcal{M}$. It applies diffusion to obtain an output image

---

[1]Note that none of the discussed methods can be truly distribution preserving since they restrict the number of possible latent sign combinations (e.g. $2^{256}$ for 256-bit PRNG seeds vs $2^{4\times64\times64}$ theoretically possible latent sign combinations. However, they can still satisfy cryptographic notions of undetectability, with the ability to yield practically identical distributions that are indistinguishable in polynomial time.

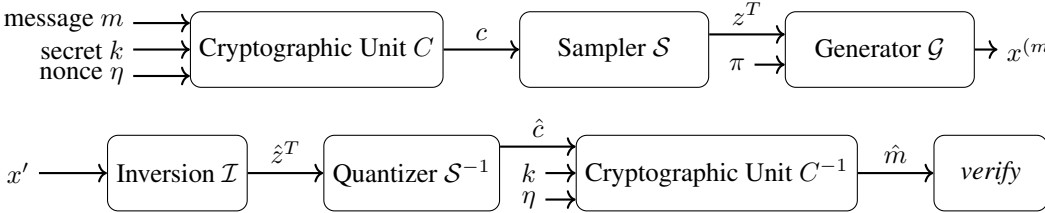

Figure 1: Diagram showing the typical generation (top) and watermark verification (bottom) procedures of a distribution-preserving inversion-based watermark.

$x$ from the initial latent $z_T$. To verify a watermark, all modules are inverted in order. Finally, a statistical check is performed to verify if $x'$ is actually watermarked and how well it matches a valid message in case of a multi-bit watermark (see Figure 1, bottom).

**Gaussian Shading** The cryptographic unit of GS takes the user id $m$ as message and replicates it such that the replicated message has the same number of elements as $z_T$. Subsequently, the replicated message is encrypted using a stream cipher as outlined above. The resulting ciphertext $c$ is then used to steer sampling of $z_T$: the sampler $\mathcal{S}$ divides the standard normal distribution in $2^\ell$ bins[2] with equal probability mass and selects bins according to the bits in $c$. For watermark verification of an image $x'$, the model provider performs a full DDIM inversion $\mathcal{I}$ to get an estimated latent noise $\hat{z}_T = \mathcal{I}(x')$. Next, the inverse sampling process $\mathcal{S}^{-1}$ is done where $\hat{z}_T$ is quantized to obtain the encrypted message bits $\hat{c}$. Afterwards, decryption and error correction are applied to recover a user id $\hat{m} = \text{Repl}^{-1}(\hat{c} \oplus \text{PRNG}(k, \eta))$. The final stage is to check if $\hat{m}$ matches with any user id $m$ known by the service provider. This is done by comparing the number of matched bits between $\hat{m}$ and every known $m$. A match is found if the number of matching bits exceeds a predefined threshold, which depends on the total amount of users/messages and guarantees sufficient statistical significance.

**Pseudo Random Codes Watermark** Pseudorandom Error-Correcting Codes (PRC, Christ & Gunn (2024)) extend LDPC codes and combine error-correction with pseudorandomness. They were specifically developed for AI watermarking and steganography. Gunn et al. (2025) adapted PRC for in-generation watermarking of diffusion models, a technique known as PRCW. When sampling a new image, a user id $m$ is encoded to a pseudorandom codeword $c$ using a PRC (as opposed to replication error-correction and stream cipher encryption in GS). Subsequently, sampling and generation are done as in Gaussian Shading. For verification, PRCW provides algorithms for both detection (is a PRCW watermark present?) and decoding (recovering the message $m$). The latter is slower since it relies on belief propagation. Both detection and decoding require inversion of the denoising process and quantization similarly to Gaussian Shading. However, PRCW uses *exact* DDIM inversion (Hong et al., 2024), which is more accurate than regular DDIM inversion used in GS, but causes a significant increase in runtime.

**Gaussian Shading++** Gaussian Shading++ (Yang et al., 2025, GS++) is an improvement over GS which is based on a fusion with PRC. Crucially, GS uses a nonce $\eta$ for the stream cipher. In GS++, this nonce is included into the watermark as the message of a PRC. So half of the latent values are used to represent a PRC which encodes just $\eta$, the other half is used for a GS watermark which uses a pseudorandom expansion (i.e. hash) of this $\eta$ and contains the actual user ID as message $m$. To verify the watermark, GS++ first decodes the PRC watermark to get back $\eta$ and uses this to extract $m$ from the GS watermark. To be able to decode the PRC, it also relies on exact inversion.

---

[2]For $\ell = 1$, we have two bins and randomly sample either a negative or a positive value $z_T[i]$ from the Gaussian distribution depending on the binary value of $c[i]$.

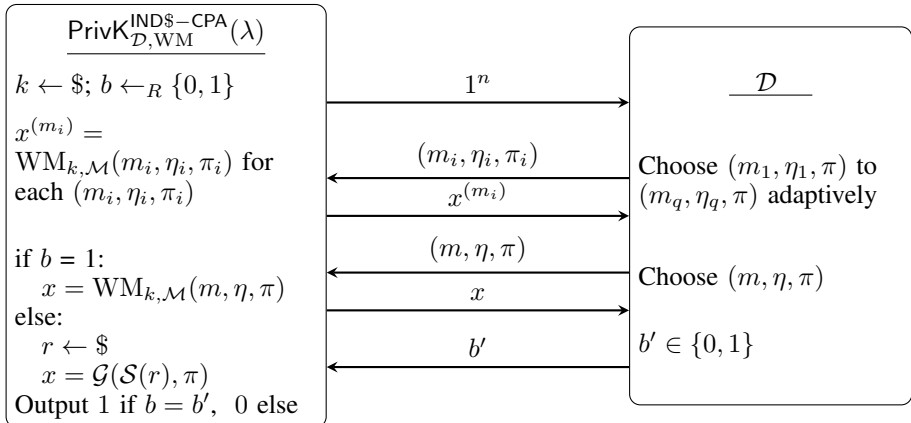

Figure 2: IND$-CPA game for a watermarking scheme WM

# 3 REVISITING CRYPTOGRAPHY FOR SEMANTIC WATERMARKING

While cryptographic primitives such as stream ciphers and PRCs can ensure pseudorandomness and thereby undetectability of the watermarking schemes, they only do so if used correctly. We provide a formal definition of undetectability for nonce-based semantic watermarks and relate it to prior work.

## 3.1 UNDETECTABILITY FOR NONCE-BASED WATERMARKING SCHEMES

In cryptography, a standard security assumption is Indistinguishability under Chosen Plaintext Attack (IND-CPA). Informally, this means that an distinguisher $\mathcal{D}$ cannot determine which message was encrypted, even if $\mathcal{D}$ knows the two possible messages and has observed encryptions of other (not necessarily different) messages before (Katz & Lindell, 2015). IND$-CPA (Rogaway, 2004) is a slightly stronger assumption, stating that $\mathcal{D}$ cannot distinguish the encryption of a known message from a random bit string without knowing the key. Formally, we define **the game** $\mathsf{PrivK}^{\mathsf{IND\$-CPA}}_{\mathcal{D},\mathrm{WM}}(\lambda)$, which is a polynomial time algorithm and depicted in Figure 2. It uses the watermarking algorithm $\mathrm{WM}_{k,\mathcal{M}}(m,\eta,\pi)$ to output a watermarked image using the secret key and a nonce which introduces randomness. In the beginning, it generates a random secret key $k$ and transmits the security parameter as $1^\lambda$ to $\mathcal{D}$.

In the **oracle phase** of the game, the distinguisher can request up to $q$ watermarked images for messages, nonces, and prompts $(m_i,\eta_i,\pi_i)$ that $\mathcal{D}$ provides. The inputs can be identical or different and for each image the game responds with the corresponding watermarked image $x^{(m_i)}$. In the **challenge phase**, the distinguisher provides a message $m$, a nonce $\eta$, and a prompt $\pi$. Based on the random bit $b$, $x$ is returned, which is either an image containing $m$ as watermark or an image generated from an initial latent $\mathcal{S}(r)$ for a distribution-preserving sampler $\mathcal{S}$ and a random seed $r$. Finally, $\mathcal{D}$ outputs a guess $b'$ if $x$ is watermarked or not, and the game checks if this is correct ($b = b'$). Overall, a watermarking scheme is called IND$-CPA-secure with a security parameter $\lambda$, if $\Pr[\mathsf{PrivK}^{\mathsf{IND\$-CPA}}_{\mathcal{D},\mathrm{WM}}(\lambda) = 1] = \frac{1}{2} + \mathrm{negl}(\lambda)$, where $\mathrm{negl}(\lambda)$ is a negligible function, i.e., $2^{-\lambda}$.

A distinguisher $\mathcal{D}$ is called **nonce-respecting** if $\mathcal{D}$ never queries the same nonce multiple times. Note that a practical distinguisher usually has no control over the nonce if a watermarking scheme is designed securely. However, in this way we can simulate different behaviour under different nonce-choosing strategies.

**Theorem 1.** *Let $C$ be an IND$-CPA secure cryptographic unit, let $\mathcal{S}$ be a distribution preserving sampler. If the distinguisher $\mathcal{D}$ is nonce-respecting, the watermarking scheme is undetectable in the IND$-CPA notion. If $\mathcal{D}'$ is not nonce respecting, it cannot be undetectable in the IND$-CPA notion.*

We sketch an outline of the proof in the following and give the full proof in Section A in the Supplementary Material. In the undetectable case, and nonce-respecting distinguisher $\mathcal{D}$ with inversion

capabilities can basically make the model transparent by inverting. Thus, it interacts directly with the cryptographic unit $C$. As this is IND\$-CPA secure, $\mathcal{D}$ cannot gain more than negligible advantage. However, if an distinguisher $\mathcal{D}'$ is not nonce-respecting, $\mathcal{D}'$ can reuse a message-nonce pair $(m, \eta)$ both in the oracle and challenge phase. In the oracle phase, it is guaranteed to observe the watermarked image $x^{(m)}$. In the challenge phase, it inverts the image $x$ and compares whether it is similar to $x^{(m)}$ or not[3]. This distinction has a very high success probability (close to one instead of negligible) and make the watermarking scheme detectable.

Note however, that it can still be distortion-free as for this a distinguisher does not get any queries in the oracle phase. However, in practice, users and thus distinguishers can generate multiple images. Our new definition holds for this case. In summary, if we do not choose a new nonce for every generated image, it is easy to distinguish these images from a random one as their latents are highly similar. However, if we alter the nonce for every image, the watermark stays hidden.

## 3.2 APPLICATION TO WATERMARKING SCHEMES

The cryptographic unit of GS relies on ChaCha20 (Bernstein et al., 2008). We assume that it is IND\$-CPA secure, as no non-generic attacks are known so far[4]. As the deployed sampler is distribution preserving, GS is clearly undetectable if the nonce $\eta$ is chosen different at each step. However, its original proof only covers the case of $q = 0$ and only proved distortion-freeness Yang et al. (2024b). Zhao et al. (2025) and Gunn et al. (2025) use a different notion of undetectability, which is broader but weaker. They build on IND-CPA (WM-IND-CPA) and consider a distinguisher $\mathcal{D}$ that has access to two image generators. One is the plain model $\mathcal{M}$. $\mathcal{D}$ needs to distinguish whether the second one is also just the plain model $\mathcal{M}$ or some watermarking scheme $\mathrm{WM}_{k,\mathcal{M}}$. We formally include this definition in Appendix B and discuss its relation to IND\$-CPA in the following.

**Theorem 2.** *Let $\mathcal{M}$ be a model with truly random latents. Then IND\$-CPA is at least as strong as WM-IND-CPA.*

The proof is provided in Appendix C. From this theorem follows that every watermarking scheme that is IND\$-CPA is also WM-IND-CPA. Christ & Gunn (2024) show that PRC is undetectable under WM-IND-CPA given suitable parameters. However, we can show that PRCW is not IND\$-CPA:

**Theorem 3.** *PRCW is not IND\$-CPA.*

The proof is based on finding a successful attack on PRCW and can be found in Appendix D. Note, that not being IND\$-CPA does not harm PRCWs undetectablity in most deployments in practice. However, based on Theorem 2 and Theorem 3 we can formulate the following theoretical peroperty:

**Corollary 1.** *IND\$-CPA is strictly stronger than WM-IND-CPA.*

Yang et al. (2025) provide a proof of IND\$-CPA for GS++. They encode the nonce of GS with a PRC. However, they shorten the nonce in PRC and use a pseudorandom expansion function on both the secret key $k$ and the nonce. This gives IND\$-CPA undetectability.

Tree-Ring like schemes (Wen et al., 2023; Ci et al., 2024; Arabi et al., 2025a) are neither undetectable nor distortion-free in any of those definitions. The main reason for this is that they alter the distribution of $z_T$ by introducing a pattern there, which is not cryptographically hidden. Therefore, they can be detected by a distinguisher $\mathcal{D}$. Nevertheless, Tree-Ring like schemes still have comparable performance in practice as the changes to the distribution are so subtle that they do not harm the image quality and variability a lot.

## 3.3 IMPLICATIONS FOR NONCE MANAGEMENT

Our novel proof of undetectability gives a blueprint of how to deploy cryptographic units and specifically, how to choose nonces. Any cipher should normally be used in a **same key, new nonce** configuration. This means the provider creates a fixed secret key $k$ once. Given a fixed message $m$ and

---

[3]Note that $\mathcal{D}'$ could even use a proxy model for inversion instead of the original model (Müller et al., 2025).

[4]Note that no real world symmetric cipher fulfils that definition in a strict sense. Nevertheless, the best known attacks require $\mathcal{O}(2^{\lambda/2})$ time (Boura & Naya-Plasencia, 2023) which we consider infeasible for any practically relevant attack. Therefore, it can only guess a key with negligible success probability or needs to observe the encryption for one of its $q$ requested nonces.

the fixed key $k$, the provider has to use a new nonce $\eta$ to control the sampling process for every generated image. In a normal message exchange setting, the sender could transmit the unencrypted nonce together with the encrypted message $c$. However, this is generally not possible in watermarking since the nonce would have to be transmitted in (removable) meta-data. An equivalent way in the semantic watermarking setup is the **new key, new nonce** configuration. However, this configuration has no cryptographic benefits compared to the previous configuration if the nonce is long enough, and only to save every key in addition. Finally, there is the **same key, same nonce** configuration. With fixed key $k$ and nonce $\eta$, the stream cipher always produces the same keystream $K$, and thus the same ciphertext $c$ if the message $m$ stays the same. This significantly impacts image variety and makes the watermark detectable.

## 4 Cryptography for Watermarks in Practice

Originally, Yang et al. (2024b) evaluated GS in the *new key, new nonce* configuration (Yang et al., Accessed: Feb. 2025) and stated key management as an open problem. In this section, we highlight the challenges associated with key and nonce management and discuss possible solutions.

In addition, subsequent publications (Gunn et al., 2025; Yang et al., 2024a; Zhao et al., 2025; Shehata et al., 2025) always assumed a *same key, same nonce* configuration, which is neither undetectable nor does it preserve image variety (as confirmed in Section E in the Supplementary Material). This fails to capture the potential of GS or similar cryptographic approaches and raises confusion in the community.

### 4.1 Key and Nonce Management

In order to enable the distribution-preserving properties of cryptography-based semantic watermarks, we must draw a new nonce for every newly generated image. The challenge is how to associate the nonce with a generated image for reliable verification. Publishing generated images can subject them to various transformations that can strip meta-data and distort the image. Thus, storing the nonce in meta-data or encoding with a steganographic approach weaker than the watermark is not a robust protection.

Another possibility is to keep track of all used nonces. This would require maintaining an internal database. However, in principle, this should not be an issue since we assume that the watermark verification service is kept private in order to prevent attacks on the watermark. In addition, generative services typically already store all user-generated images on their servers so the cost of storing nonces is negligible in comparison. This is the solution we further explore in this section, starting with a naive baseline as implied by Yang et al. (2024b), and presenting other, more efficient solutions.

**Baseline** As a baseline ("GS Reference"), we consider the case where all user ids and nonces are stored separately. In this case, in order to retrieve the user id, the verification procedure must iterate over all nonces and try to decrypt the message. Subsequently, since by default we have no way of knowing that the decrypted message is valid (that the nonce was correct), we must iterate over all user ids to find the closest matching one. If we consider $N$ users which generated $M$ images in total, it runs in time $\mathcal{O}(MN)$[5].

**Randomness check** In the baseline, we search over all user ids for every nonce we try. However, this is not necessary since we can exploit the repetition code of GS to check how likely it is that the matching nonce was found. We search over the user ids only for the most promising nonces. Concretely, given the recovered encrypted bitstring $\hat{c}$, a candidate decryption $s$ is obtained by decrypting with a candidate nonce $\eta'$. If $\eta'$ was correct we $s$ is encoded message, if not a pseudorandom bitstring. We perform a statistical test (see Appendix F) to determine whether $s$ is likely to be an encoded message and only then decode and search the user ids. This method ("GS Rand Check") lowers the runtime to $\mathcal{O}(M + N)$ and doesn't require any additional stored data structures.

---

[5]Note that it is possible to speed up user id search using advanced retrieval techniques, like LSH.

**Database lookup**   Another optimization of the user id lookup procedure would be to use a database to associate the nonces to the corresponding user ids (recall that every generated image has its own nonce and is generated by exactly one user). Verification still requires iterating over all nonces, until a nonce is found for which the associated user id matches the user id recovered from the image. This ("GS DB Lookup") lowers the complexity to $\mathcal{O}(M)$, but at the cost of $\mathcal{O}(M)$ additional data.

**Sorting by perceptual hashes**   All previous methods require searching through all the nonces linearly. This can be improved by using feature-based image lookup, which should greatly speed-up nonce retrieval for large-scale collections. In our experiments, we use DINO (Caron et al., 2021) to compute image feature for every generated image and store these features in a vector database (we use Chroma[6]). User ids and nonces are stored alongside DINO features. During verification, the DINO features of the incoming image are computed, and used to retrieve database entries with the most feature vectors. Starting from most similar entry, we decode with the stored nonce, and verify whether the stored user id matches the decrypted id. This method ("GS Chroma") runs in $\mathcal{O}(\log M + k)$, with $\mathcal{O}(M)$ additional entries in a database.

### 4.2 EXPERIMENTAL SETUP

We set the number of images $M = 10,000$ and number of users $N = 1,000$, where every user creates images according to a normal distribution with mean 10 and $\sigma = 3$. We measured the performance of the watermarking schemes in terms of bit accuracy and time. We measure both the runtime of the cryptographic part as well as the runtime of the inversion, since PRC and GS++ depend on exact inversion, which is much more costly.

We measure the runtime in three scenarios to accommodate three typical cases when a semantic watermark is deployed. First, we consider the case where nonces are iterated in the best possible order. This case is relevant if a service provider sorts the nonces according to the creation date to account for the probability that recently created images are also more likely to be checked. We define this case as *best case*. We consider finding it within the first 100 tries. Second, we consider the *worst case*. This precisely means that the nonce was not found, since the image was not watermarked by the service, as it e.g. was not generated by it. In this case, we always check all nonces and should not find a matching user id. Third, we consider the *average case*. We did this by selecting the nonces to be equally spaced within the $M$ nonces we considered. We used Stable Diffusion 2.1 (Rombach et al., 2022) to ensure comparability to previous work (Yang et al., 2024b; Gunn et al., 2025; Yang et al., 2025). Details on the computing infrastructure and exact software versions can be found in Section G in the Supplementary Material.

### 4.3 RESULTS

| | TPR | | | Verification Time (s) | | | Inversion + Verification Time (s) | | |
|---|---|---|---|---|---|---|---|---|---|
| | Best | Avg. | Worst | Best | Avg. | Worst | Best | Avg. | Worst |
| GS++ | 0.95 | 0.61 | 0.00 | 0.04 | 0.06 | 0.04 | 85.07 | 85.09 | 85.07 |
| PRCW | 1.00 | 1.00 | 0.00 | 0.45 | 0.45 | 0.45 | 85.48 | 85.47 | 85.48 |
| GS Reference | 1.00 | 1.00 | 0.00 | 41.14 | 40.13 | 40.06 | 46.14 | 45.12 | 45.05 |
| GS Rand Check | 1.00 | 1.00 | 0.00 | 0.03 | 2.73 | 4.70 | 5.02 | 7.72 | 9.70 |
| GS DB Lookup | 1.00 | 1.00 | 0.00 | 0.01 | 1.21 | 2.46 | 5.01 | 6.20 | 7.45 |
| GS Chroma | 1.00 | 1.00 | 0.00 | 3.38 | 4.45 | 3.38 | 8.38 | 9.44 | 8.38 |

Table 1: Performance comparison of watermark verification methods.

Without noise, most schemes perform fairly well in terms of accuracy, as shown in Table 1. However, there are differences with respect to time. The verification of GS Reference always takes worst case time as it always searches all the user ids. This is not applicable for larger scale, but can be mitigated to some extent by our proposed improvements. While PRCW is not the fastest, it is constant time,

---

[6]Chroma builds on SQLite3 but supports $k$ approximate nearest neighbour search on vector embeddings with *Hierarchical Navigable Small Worlds* (Malkov & Yashunin, 2020). Code is available at `https://github.com/chroma-core/chroma`

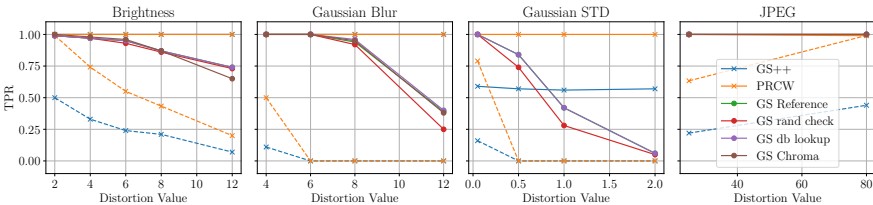

Figure 3: Robustness of watermarking schemes. The recommended inverison is solid. In case of GS++ and PRCW, we also included DDIM inversion as dashed.

which is a very desirable property in large scale applications. However, GS++ is even faster due to lower complexity of the involved ECCs. Nevertheless, there is a trade-off involved: PRCW and GS++ depend on exact inversion, which is magnitudes slower than DDIM inversion, such that their runtime was entirely dominated by it. If we assess the robustness with the default choice of inversion technique (see Figure 3, we observe that the robustness is dominated by the inversion method. While all schemes are tolerant to a small amount of noise, DDIM inversion fails to provide accurate latents for larger noise levels. However, if we compare schemes in terms of DDIM inverison only (dashed lines for PRCW and GS++), we see that those schemes are in fact less robust than GS.

We conclude that there is a trade-off. PRCW and GS++ provide a slow but constant-time verification, which is very robust to perturbations because of an expensive inversion routine. GS on the other hand can be deployed with faster DDIM inversion and has good performance if only a medium amount of images needs to be produced.

## 5 RELATED WORK

Watermarking for images has been studied for decades now (Cox et al., 2002; 2007; Al-Haj, 2007; Zhang et al., 2019).More recently, watermarking specifically for generative AI has been studied (Huang et al., 2024; Song et al., 2021; Feng et al., 2024). SEAL Watermark Arabi et al. (2025b) is a single bit watermark which uses perceptual hashes to include the image content. CLUE watermark Shehata et al. (2025) is a single-bit watermark which directly builds on a hard cryptographic problem and is a continuous extension to PRCs. However, its generalisation to a multi-bit setting is unclear. There exist Gaussian Shading derivates for tabular data generation Zhu et al. (2025) and for video generation Hu et al. (2025), to which our results extend as well. Aremu et al. (2025); Yang et al. (2024a) experimentally attacked watermarks by exploiting patterns. They provide empirical evidence that same key, same nonce watermarking schemes are detectable, but did not provide theoretical insights.

For LLMs, there exist different constructions of undetectable watermarks. Generally, they bias the token distribution in an undetectable way. Christ et al. (2024) were the first to achieve this based on cryptographic one way functions, which was later extended by Fairoze et al. (2025). PRC Christ & Gunn (2024) also includes usage as an undetectable watermarking scheme for LLMs.

## 6 CONCLUSION

We provide a framework for the general understanding of cryptography-based semantic watermarks. We further establish undetectability under IND$-CPA for watermarking that explicitly considers the use of randomness. We show that this is crucial for the undetectability of schemes like Gaussian Shading and Gaussian Shading++. We make explicit that GS—while designed as an undetectable watermark—was not used as such in subsequent literature because of a different deployment mode. Further we relate IND$-CPA to existing literature and find that it is an even stronger notion of undetectability. We propose lightweight modifications to Gaussian Shading to make it still applicable in practise and compare this to existing schemes. GS can be used with a faster and less robust DDIM inversion. With our modifications, we also achieve runtime advantages which makes it suitable for small scale deployment schemes. However, for large scale deployments, PRCW and GS++ are a better choice as they provide constant time watermark verification.

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

## A   PROOF OF THEOREM 1

We show that if the sampler is distribution preserving, a distinguisher $\mathcal{D}$ is nonce respecting and the watermarking scheme uses an IND\$-CPA secure cryptographic unit $C$, the watermarking scheme is undetectable under IND\$-CPA.

Formally, we get

$$\Pr[\mathsf{PrivK}_{\mathcal{D},\mathrm{WM}}^{\mathsf{IND\$-CPA}}(\lambda) = 1] = \Pr[b = 1]\Pr[\mathcal{D}(x) = 1] + \Pr[b = 0]\Pr[\mathcal{D}(x) = 0] \tag{1}$$

$$= \frac{1}{2}\Pr[\mathcal{D}(\mathrm{WM}_{k,\mathcal{M}}(m,\eta,\pi)) = 1] + \frac{1}{2}\Pr[\mathcal{D}(\mathcal{G}(\mathcal{S}(r))) = 0] \ . \tag{2}$$

On a real random input (second term), $\mathcal{D}$ cannot obtain any information and therefore just guesses with probability $\frac{1}{2}$. On a watermarked input (first term), $\mathcal{D}$ needs to recognize the output of IND\$-CPA secure cryptographic unit for an unknown key, which is hard by assumption. Therefore, the right hand side gets

$$= \frac{1}{2}(\frac{1}{2} + (q+1)\operatorname{negl}(\lambda)) + \frac{1}{2} \cdot \frac{1}{2} = \frac{1}{2} + \operatorname{negl}(n) \ . \tag{3}$$

If we consider a distinguisher $\mathcal{D}'$ that is **not nonce-respecting**, there is an obvious attack. First, $\mathcal{D}'$ chooses $m^\star$, $\eta^\star$, and $\pi^\star$ and passes this as $(m_1, \eta_1, \pi_1)$ and as $(m, \eta, \pi)$. $\mathcal{D}'$ obtains $x^{(m_1)}$ and $x$. Next, $\mathcal{D}'$ uses inversion and the inverse sampler to recover the ciphertexts $c_1 = \mathcal{S}^{-1}(\mathcal{I}(x^{(m_1)}))$ and $c = \mathcal{S}^{-1}(\mathcal{I}(x))$. Idealized, if they both match, $\mathcal{D}'$ has found that this image is watermarked and outputs 1, otherwise 0. Usually, they will not be exactly the same due to error in the recovery. However, they are close enough such that recovery is possible, i.e., $c \approx c_1$. We compute the probability for the distinguisher $\mathcal{D}'$ and find that

$$\Pr[\mathsf{PrivK}_{\mathcal{D}',\mathrm{WM}}^{\mathsf{IND\$-CPA}}(\lambda) = 1] = \Pr[b = 1]\Pr[\mathcal{D}'(x) = 1] + \Pr[b = 0]\Pr[\mathcal{D}'(x) = 0] \tag{4}$$

$$= \frac{1}{2}\Pr[\mathcal{D}'(\mathrm{WM}_{k,\mathcal{M}}(m,\eta,\pi)) = 1] + \frac{1}{2}\Pr[\mathcal{D}'(\mathcal{G}(\mathcal{S}(r))) = 0] \tag{5}$$

$$= \frac{1}{2} \cdot 1 + \frac{1}{2}(1 - \operatorname{negl}(n)) \tag{6}$$

$$= 1 - \frac{1}{2}\operatorname{negl}(n) \ . \tag{7}$$

Clearly, $\mathcal{D}'$ has a non-negligible success probability—which is in fact close to 1 even with just one watermarked image—and can therefore easily distinguish between an unwatermarked image and a watermarked one.

## B   GENERAL DEFINITION OF DISTRIBUTION-PRESERVING WATERMARKS

We provide a game based definition of WM-IND-CPA, which equivalent to the is the undetectability notion used by Zhao et al. (2025) and Gunn et al. (2025).

The game is depicted in Figure 4 and works as follows: A distinguisher is either given access to a plain image generation model $\mathcal{M}$ or to a watermarking scheme $\mathrm{WM}_{k,\mathcal{M}}$ depending on a coin flip by the game. It can request up to $q$ arbitrary adaptive messages and prompts. If it has access to $\mathcal{M}$, the messages are ignored. WM is called undetectable if there does not exist a distinguisher that can win the game $\mathsf{PrivK}_{\mathcal{D},\mathrm{WM}}^{\mathsf{WM-IND-CPA}}(\lambda)$ with more than negligible advantage. Formally, we require that for every polynomial time distinguisher $\mathcal{D}$, $\Pr[\mathsf{PrivK}_{\mathcal{D},\mathrm{WM}}^{\mathsf{WM-IND-CPA}}(\lambda)] = \frac{1}{2} + \operatorname{negl}(\lambda)$.

This definition is an equivalent formulation of Zhao et al. (2025).

While this definition is more generic, is does not explicitly expose the nonce of a watermarking scheme to our analysis, which is why we choose to introduce nonce-based IND\$-CPA.

## C   REDUCTION FROM IND\$-CPA TO WM-IND-CPA UNDETECTABILITY

We now consider the relation between WM-IND-CPA (see Appendix B) and IND\$-CPA.

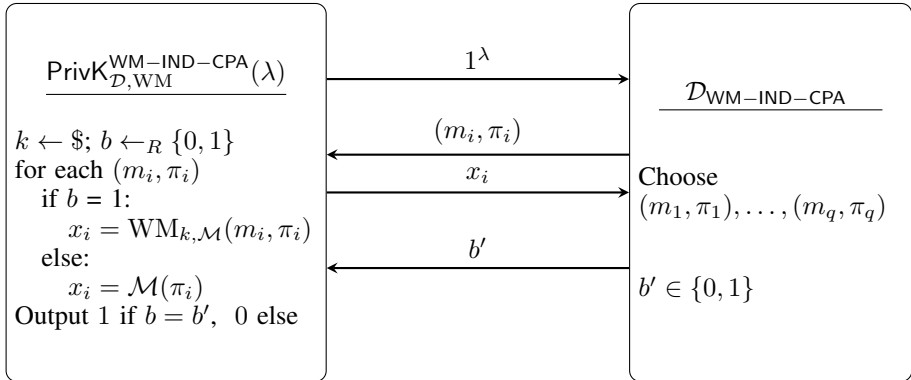

Figure 4: WM-IND-CPA undetectability.

We show that IND$-CPA is at least as strong as the WM-IND-CPA. We assume that the unwatermarked model actually uses truly random initial latents $z_T$. Otherwise all watermarking schemes proposed so far are delectable. Under this assumption, distinguishing outputs from $\mathcal{M}$ is equivalent to distinguishing images from random seeds (see "else case" in Figure 2). We start with identifying the WM-IND-CPA with $q$-IND$-CPA by making the following observation. It does not matter whether a distinguisher which can request an arbitrary polynomial amount of images requests these upfront in the oracle phase or all of them in the query phase. It is equivalent to a distinguisher that requests up to $2q$ images.

It is a well known fact in cryptography, that such a distinguisher has an advantage of $2q$ times over a distinguisher which just has access to one image in the oracle phase Katz & Lindell (2015). We achieved the canonical IND$-CPA game which does not require nonces (we write c-IND$-CPA). This undetectability notion does contain an oracle phase to query watermarked images and a challenge phase in which a distinguisher needs to tell for one challenge image whether it was watermarked or not.

The final step reduces our nonce-based undetectability to c-IND$-CPA. This is straight forward. All the reduction needs to do is for every query simulate a nonce respecting choice of nonces. We assume a strategy that chooses them distinct by some deterministic scheduling. In this case, the reduction perfectly simulates the c-IND$-CPA such that the advantage transfers.

To summarise, we have shown that nonce-based IND$-CPA undetectability is at least as strong as WM-IND-CPA undetectability under the assumption that the model actually chooses perfectly random initial seeds.

## D  PRCW IS NOT IND$-CPA UNDETECTABLE

Next, we are going to show that PRCW is not undetectable under nonce-based IND$-CPA. More precisely we show that a nonce-respecting distinguisher $\mathcal{D}$ can break the pseudorandomness if it chooses nonces distinct but non-random.

In order to do so, we first provide a more detailed definition of a PRCW code word. Then, we provide an attack on PRCW as proof. Finally, we outline its consequences on both PRCW and GS++.

**PRCW**  A PRC extends a normal LDPC code first by encoding a concatenation of a message $m$, some check bits $\beta$ and a nonce $\eta$. The check bits are used to ensure a desired FPR on detecting watermarks in general. They are always the same and matched upon watermark verification. The nonce ensures that if the same $m$ is encoded twice, its values differ. Furthermore, it contains a sparse binary error vector $e \sim Bin(|c|, \delta)$ for a small $\delta$, which ensures that recovering $G$ from some $c$ is computationally hard. Finally, it contains a random but fixed bit string otp of length $|c|$ that

ensures a technical independence property. As an encoding, we get the following equation over bits. $c = G(\beta||m||\eta)^T \oplus e \oplus \text{otp}$.

**Proof of Theorem 3**   The attack works as follows: First, $\mathcal{D}$ requests code words for a fixed message $m$ (e.g. $m = 0$) and for four nonces $\eta_1, \ldots, \eta_4$ with the property that $\bigoplus_{i=1}^4 \eta_i = 0$. It requests the first three in the first phase to obtain the valid PRC code words for these and the last one in phase two. Afterwards, $\mathcal{D}$ XORs all the received codewords $c_i$. If all four of them are code words, the resulting bitstring should only contain a small amount of ones. Otherwise, it should be fairly large.

In more detail, we assume that $b = 1$ and therefore all four code words are valid. We end up with the XOR of the noise vectors $e_i$. Not that for any constant, the XOR of an even number of it is zero, e.g. $\bigoplus_{i=1}^4 \text{otp} = 0$.

$$\bigoplus_{i=1}^4 c_i = \bigoplus_{i=1}^4 G(\beta|\eta_i|m)^T \oplus e_i \oplus \text{otp}$$

$$= G \left( \bigoplus_{i=1}^4 (\beta|\eta_i|m)^T \right) \oplus \bigoplus_{i=1}^4 e_i$$

$$= G \cdot 0 \oplus \bigoplus_{i=1}^4 e_i = \bigoplus_{i=1}^4 e_i$$

If code word is random, we expect about half of the bits to be one. If not, it should be the XOR of four independent noise vectors from $Bin(|c|, \delta)$, which contain less than $4|c|\delta$ ones on expectation, which far from $\frac{n}{2}$. How far depends on the specific choice of parameters for $\delta$ and $|c|$, however a possible configuration for our experimental section (see section 4.2 is $|c| = 2^{14} = 16,384$ and $\delta = 0.0081$. It is obvious to see that the probability of $\mathcal{D}$ being able to distinguish is far from negligible and more close to one. $\qquad \square$

**Consequences for PRCW and GS++**   Our attack strategy is not valid if $\mathcal{D}$ chooses nonces uniformly at random, which is how PRCW was proposed by Gunn et al. (2025). In this case, the best strategy is a meet in the middle attack, but the nonce length of a PRC is chosen such that it is infeasible in accordance with the security parameter $\lambda$. As a consequence, PRCW is undetectable if nonces $\eta$ are chosen uniformly at random every time, but does not fulfil the stronger notion of IND$-CPA security.

This extends to GS++. There, a PRC is used but without any form of check bits $\beta$, additional nonce $\eta$ or $\text{otp}$. Instead, its encoded message it the nonce of the entire scheme. However, this nonce is shorter. It is used as a salt in a hash function together with the secret key $k$ to expand it to a full pair of stream cipher key and nonce. If at this point a pseudorandom expansion function like SHAKE-128 (Bertoni et al., 2009) is used, the nonce is actually pseudorandom at each step. This gives IND$-CPA.

# E   IMPACT OF WATERMARK CONFIGURATIONS ON IMAGE DIVERSITY

To show the impact of different watermarking schemes and their configurations on image diversity, we evaluate two metrics empirically. First, we calculate the FID (Heusel et al., 2017) between five non-overlapping sets of 2.000 watermarked and unwatermarked images taken from a total of 10.000 images each and report mean and standard deviation. Generation prompts are taken randomly from the Stable Diffusion Prompts dataset[7] and are aligned in each set, meaning images in both sets are generated from the same list of prompts. Second, we calculate the mean pairwise LPIPS (Zhang et al., 2018) scores between pairs of images within sets of 100 images 10 times, generated from 10 different prompts and report mean and standard deviation from the 10 runs. Like in the main experiments, we use Stable Diffusion V2.1, the DPM sampler[8], as well as default guidance scale (7.5) and inference steps (50).

---

[7]Stable Diffusion Prompts

[8]DPMSolverMultistepScheduler

**Results** All watermarking schemes show FID and LPIPS scores comparable to unwatermarked generation with the exception of Gaussian Shading setup with same keys and same nonces. For this setup, the FID scores are higher and LPIPS scores are lower, indicating more similar outputs which is due to similar initial latents

| | FID ↓ | LPIPS ↑ |
|---|---|---|
| No Watermark | $25.1281_{\pm 0.3018}$ | $0.5921_{\pm 0.0333}$ |
| GS Reference | $25.0959_{\pm 0.2396}$ | $0.5946_{\pm 0.0315}$ |
| GS - New key, new nonce | $25.0658_{\pm 0.2101}$ | $0.5945_{\pm 0.0318}$ |
| GS - Same key, same nonce | $25.4393_{\pm 0.3335}$ | $0.5861_{\pm 0.0321}$ |
| GS++ | $25.0975_{\pm 0.1872}$ | $0.5932_{\pm 0.0313}$ |
| PRCW | $25.1470_{\pm 0.1485}$ | $0.5950_{\pm 0.0319}$ |

Table 2: Experiments on image diversity under different watermarking schemes and settings.

## F RANDOMNESS CHECK

We start with the following observation: by inversion, we recover a long bitstring $\hat{c}$. If it gets decrypted with a wrong nonce, we will obtain a pseudorandom bitstring. This is the case because $\mathrm{PRNG}(k, \eta')$ is a new pseudorandom sequence, so $\tilde{s} = \hat{c} \oplus \mathrm{PRNG}(k, \eta')$ is a new encryption of $\hat{c}$ instead of a decryption. Since our stream cipher is secure, this is pseudorandom. Hence, the amount of ones in each strip of length $\rho$ is binomially distributed according to $Bin(\rho, \frac{1}{2})$. If we chose the correct nonce, we will get a repetition code where each bit is repeated $\rho$ times (see Section 2.2) up to some error.

We therefore observe the amount of ones in each block of $\rho$ consecutive bits and determine the probability, that this was generated by a binomial distribution with $Bin(\rho, \frac{1}{2})$ (random bits) or by a biased one (potentially an encoded bit). We choose a standard test power of 80%. There are $|m|$ blocks in $\tilde{s}$. If it was indeed random, we expect each individual test to fail with a probability of 20%. We examine whether more tests failed than one would expect from $|m|$ samples from $Bin(\rho, \frac{1}{2})$ by performing an additional one-sided test over all. We set our FPR to 0.01, since false positives are costly in runtime, but false negatives would imply a missed detection. If we consider the $\tilde{s}$ non-random, we assume that we actually found the correct nonce $\eta$ and continue with the verification. This gives the runtime of $\mathcal{O}(M + N)$.

Our method involves a trade-off: If we choose FPR too small, we essentially lower the error correction capabilities of the underlying repetition code and make GS less robust. However, if we choose FPR too large, we have higher runtime. We empirically found 0.01 a valid choice.

## G FULL EXPERIMENTAL DETAILS

All experiments were conducted on a server with 4 Nvidia A40 GPUs, 32 AMD EPYC 7282 16-Core Processors, and 500 GB RAM, running Ubuntu 20.04.6 LTS. The exact Python version and package requirements can be found in our project repository (https://anonymous.4open.science/r/submission-F475/).

Across all experiments, the image sizes and the latent sizes are set to $512 \times 512$ and $4 \times 64 \times 64$, respectively.

For all watermarking schemes, we set a message length of 256 and false positive rate of $10^{-6}$.

The exact parameters used for individual schemes are as follows:

- **PRCW**
    - sparsity $t$: 3
    - error probability $\delta$: 0.0081
    - length of nonce $\eta$: 39 bits

- length of check bits $\beta$: 30 bits
    - Decoding iterations: 8
- **GS++**
    - sparsity $t$: 7
    - error probability $\delta$: 0.0176
    - length of nonce $\eta$ (ChaCha): 96 bits
    - length of nonce in PRC (PRC): 256 bits
    - Hash function: SHAKE-128
- **GS**
    - Key length: 256 bits
    - Nonce length: 96 bits

We furthermore set both denoising steps and inversion steps to 50, both for regular DDIM inversion as well as exact inversion.

All prompts used for experiments are taken from the Stable Diffusion Prompts dataset[9].

## H EXAMPLE IMAGES

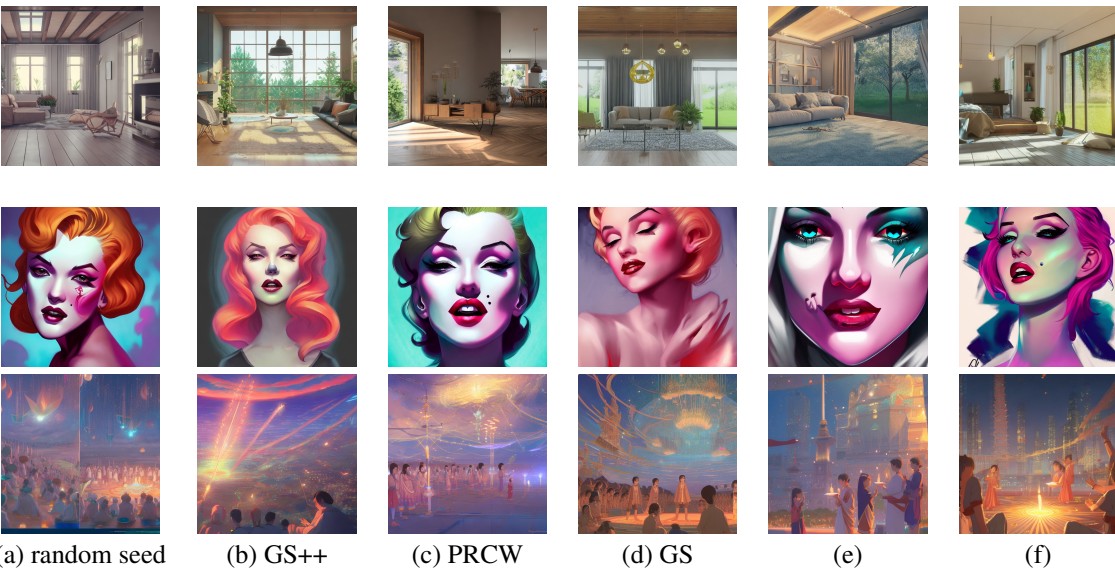

(a) random seed      (b) GS++      (c) PRCW      (d) GS      (e)      (f)

Figure 5: Comparison of different watermarking schemes. (e) is GS in the same key, same nonce setting. (f) is GS in the new key, new nonce setting.

---

[9]Stable Diffusion Prompts

