# OpenReview forum: "Cryptography in Semantic Watermarks: Undetectability and Deployment Implications"
_ICLR.cc/2026/Conference — ICLR 2026 Conference Withdrawn Submission_

### Official Review · Reviewer_zAGn · 2025-10-20

**Soundness:** 3
**Presentation:** 2
**Contribution:** 1
**Rating:** 2
**Confidence:** 4

**Summary:**

The paper provides new results related to the cryptographic approach to watermarking large language models (LLMs). Informally, a watermarking scheme relies on a secret key in order to watermark the outputs generated using the model; importantly, the watermarked outputs should be indistinguishable from non-watermarked once, a property referred to as undetectability. Additional properties required by watermarking schemes are soundness (i.e., outputs produced independently of the model should not be detected as watermarked), and robustness (i.e., even if the watermarked outputs are partially tampered with, it should be hard to remove the watermark).

The main results of the paper are concerned with the definition of undetectability for a special class of watermarking schemes known as distribution-preserving semantic watermarks. Such schemes employ cryptographic techniques to generate pseudorandom bitstrings which in turn can be exploited to sample the initial latent from the original unwatermarked initial latent distribution (typically Gaussian). This suffices to proving undetectability against computationally-bounded distinguishers.

The first contribution is a new definition of undetectability dubbed IND{\$}-CPA. This new definition basically allows the attacker to specify the random nonce used by the watermarking scheme (along with a message and a prompt). The only restriction is that all the nonces that are queried are distinct. The authors then show that their notion can be obtained generically by combining any standard IND{\$}-CPA stream cipher with a distribution-preserving sampler, as done in the scheme Gaussian Shading by Yang et al. Their analysis uncovers the necessity of using random nonces during watermarks generation.

Finally, the authors show that IND{\$}-CPA is strictly stronger than the related notion of WM-IND-CPA proposed by Christ and Gunn and achieved by watermarking schemes based on so-called pseudorandom error-correcting codes (PRCs). They conclude the paper by a series of remarks on nonce and key management for watermarking schemes.

**Strengths:**

The central claims in the paper are adequately supported with evidence. Theoretical statements are supported by proofs that are deferred to the appendix; as far as I could see, the proofs are correct.

The experimental setup compares different solutions in terms of nonce management. In the naïve, baseline, approach, the authors store a different nonce for every user id and decrypt trying all possible nonces; this is rather inefficient, and only applicable to logarithmic user ids. In the approach based on randomness check, the authors exploit the repetition code of Gaussian Shading in order to check how likely it is that the matching nonce was found; this still requires an exhaustive search over the user id space. In the approach based on perpetual hashes, the authors use DINO to compute an image feature for every generated image and store these features in a vector database. I found this solution to be unclear and not explained properly; for instance, the complexity is O(log(M) + k) but the parameter k is undefined.

I think the presentation has a large margin of improvement. Many concepts, especially those related to watermarking and LLMs, are introduced without any intuition of explanation. The authors are encouraged to improve the presentation by explaining more clearly, and in more details, non-standard concepts such as inversion-based semantic watermark or Gaussian Shading. Those are currently very hard to follow for non-experts.

While the topic is timely, the contributions are rather narrow and follow readily from well-known facts in the field of cryptography. First off, it remains unclear why the new definition is useful; even if the definition is stronger than WM-IND-CPA, the authors explicitly say that the latter is enough for most applications.
Second, the fact that stream ciphers should use random nonces is well-known. So, it is not surprising some schemes fall apart in case of nonce repetitions.
For these reasons, I don’t think the results are valuable to share with the broader ICLR community.

In light of the above, the paper does not have many strengths in my opinion. The new definition is not well motivated and is not supported by providing a killing application. The insights on key and nonce management are not new, and the experimental evaluation adds very little to them.

**Weaknesses:**

As already hinted above, the main weakness of the paper is the lack of a supporting application for the newly introduced notion. Another weakness stems from the fact that the comparison with prior work is incomplete. For instance, the authors prove that Gaussian Shading instantiated with strong-enough stream ciphers achieves their IND$-CPA notion, while watermarking based on PRCs does not. However, the latter provably satisfies the properties of soundness and robustness, and I was unable to verify this also holds for Gaussian Shading.

**Questions:**

*) Does Gaussian Shading provably satisfy the properties of soundness and robustness, as defined by Christ and Gunn?

*) Does Gaussian Shading achieve undetectability in the presence of chosen-ciphertext attacks, as defined by Alrabiah et al. (STOC 2025)? Would it be enough to use an IND-CCA secure stream cipher as the underlying cryptographic unit?

Omar Alrabiah, Prabhanjan Ananth, Miranda Christ, Yevgeniy Dodis, Sam Gunn: Ideal Pseudorandom Codes. STOC 2025: 1638-1647.

*) Why do you need to store the nonces? Can’t the nonces be randomly-generated and included in the ciphertext (not in the metadata)? I don’t think the nonces should be part of the definition at all; encryption and watermarking schemes can be randomized.

---

### Official Review · Reviewer_msF7 · 2025-10-22

**Soundness:** 3
**Presentation:** 1
**Contribution:** 3
**Rating:** 6
**Confidence:** 2

**Summary:**

This paper highlights flaws in the existing usage of cryptography-based in-process watermarks like Gaussian Shading or PRCW. They find that the associated papers lack cryptographic rigour and improperly use randomness in either an inefficient or insecure way. To resolve these issues, they propose a cryptographic framework to frame these works and prove their security, assuming proper use of randomness. Then, they study how different key/nonce management systems can affect performance in terms of watermark detection and runtime.

**Strengths:**

1. The paper presents a strong argument that existing methods lack clarity for practical use and proposes a solid cryptographic foundation to address this issue.
2. The background and related work are well explained, and the motivations of the paper are clear.

**Weaknesses:**

Major:
1. The paper is littered with typos, writing mistakes, and confusing writing. Here are some examples of the most egregious ones: “within every inversion step. reduces the added reconstruction noise.” (Line 91), “If η′was correct we sis encoded message, if not a pseudorandom bit-string” (Line 373). The paper needs to be thoroughly proofread.
2. In the results, only the TPR is provided to measure watermark detection performance. TPR alone is not enough, as a method that always answers “Yes” will have a TPR of 100% but a very high FPR. The FPR used should be clearly included and justified. Without it, the TPR results are meaningless.

Minor:
1. In Table 1, for GS Reference, the verification time of the best case scenario is worse than that of the average case, which is worse than that of the worst case. The results should include confidence intervals.
2. Some of the baselines feel pointless, although their presence does not particularly harm the paper. While GS DB lookup technically requires more storage than GS Reference or GS Rand Check, you mention that practitioners already store the generated images, making the additional storage cost of the lookup table irrelevant compared to the storage cost of the images. Your section is called “Cryptography for Watermarks in Practice” but GS Reference and GS Rand Check do not seem very practical.
3. The GS Chroma baseline is vulnerable to adversarial example attacks, which aim to target worst-case retrieval runtimes by modifying the DINO features to be as far as possible from the original features. This vulnerability is worth considering because this weakness is unique to this baseline.
4. No LLM usage statement.

**Questions:**

1. In Table 1, for the GS Reference, the verification time of the best-case scenario is worse than that of the average case. Is this due to inherent randomness in the runtime? If not, then what causes it?
2. In your results, what ratio of watermarked vs non-watermarked images did you use for the average case? How do you justify that choice?
3. At the end of your conclusion, you claim that your modifications make GS suitable for small-scale deployment, while for large-scale deployment, GS++ or PRCW should be used instead, since they are constant-time algorithms. An empirical experiment where you increase the number of images M and users N would have been helpful to reinforce your point, or an analysis of the constants in the runtime would have been beneficial. In practice, your modified GS may remain more efficient for even large and realistic deployments.

---

### Official Review · Reviewer_zrpY · 2025-10-31

**Soundness:** 2
**Presentation:** 2
**Contribution:** 1
**Rating:** 2
**Confidence:** 4

**Summary:**

The paper examines the cryptographic foundations of some semantic watermarking schemes for latent diffusion models. In the work, it is proposed to use the undetectability notion based on a known IND$-CPA scheme.

**Strengths:**

1) The authors confirm that the reuse of nonce reuse may have implications for real-world deployments: considered recent works implicitly assume a “same key, same nonce” setting, which this work shows leads to detectable watermarks.
2) The authors propose a method to fix the use of the randomness in GS without introducing a significant computational overhead.

**Weaknesses:**

1) The novelty of the paper is limited: it mainly shows the importance of nonce-respecting usage, which has long been known in cryptography; the notion of undetectability is based on a known notion for stream ciphers (IND$-CPA).

2) The robustness evaluation is poor (Figure 3), since it only includes simple images perturbations (brightness adjustment, blur, additive noise, JPEG). Missing geometric attacks (cropping, rotation, scaling), learning-based removal attacks (e.g., fine-tuning or diffusion-based watermark stripping), overwriting attacks and adaptive attacks should be added. Without these, claims about “robustness” are incomplete.

3) The paper misses important quantitative metric of the considered watermarking schemes: false positive rates, storage overheads for the millions of generated images.

4) The writing should be improved.
Overall, the scientific novelty of the work does not seem to be enough.

**Questions:**

See weaknesses.

---

### Official Review · Reviewer_bG4M · 2025-11-01

**Soundness:** 3
**Presentation:** 3
**Contribution:** 3
**Rating:** 6
**Confidence:** 1

**Summary:**

This paper investigates "distribution-preserving" semantic watermarks for latent diffusion models (LDMs), such as Gaussian Shading (GS) and Pseudo-random Codes Watermarks (PRCW). These methods rely on cryptographic primitives to ensure "undetectability." The authors find that the cryptographic foundations of these methods, particularly regarding the use of randomness (nonces), have been flawed or ambiguous in prior work.

The paper's core contribution is the introduction of a formal **IND$-CPA security framework** to rigorously define undetectability for these watermarks. This framework reveals a critical insight: **reusing a nonce** (i.e., the "same key, same nonce" configuration) makes the watermark **trivially detectable**. The paper establishes that the only cryptographically sound deployment is a "same key, new nonce" configuration for every generated image.

This insight, however, uncovers a major practical deployment problem: if every image has a unique nonce, how can a verifier efficiently find the correct nonce to check the watermark? A naive search would require $\mathcal{O}(MN)$ time (for $M$ images and $N$ users), which is infeasible.

To solve this, the paper proposes and evaluates several practical speed-up strategies for GS, such as statistical checks ("GS Rand Check") and database-backed retrieval ("GS DB Lookup", "GS Chroma"), successfully reducing the verification time to $\mathcal{O}(\log M + k)$. The work concludes with a comprehensive experimental comparison of these optimized GS schemes against PRCW and GS++ in terms of speed, robustness, and quality.

**Strengths:**

-  The primary strength is the formalization of "undetectability" using the **IND$-CPA** security framework. This brings much-needed cryptographic rigor to the field, clarifying ambiguities and providing a solid foundation for future work on distribution-preserving watermarks.

- This work excels by connecting theory to practice. It does not just identify a theoretical flaw (nonce reuse) but immediately addresses the severe *engineering challenges* (the $\mathcal{O}(MN)$ verification cost) that the "correct" solution creates.

- The evaluation in Section 4.3 and 5 is comprehensive and solid.

**Weaknesses:**

The whole paper looks good to me.

**Questions:**

All the proposed GS acceleration schemes rely on a critical assumption: the verifier has access to a private, server-side database containing all nonces and/or corresponding image features. This model is only valid for closed-source, API-based services. Is there any possible variants for the open-weights model scenario, where the goal is third-party or public verification?

---

### Note · Authors · 2025-11-28

**Comment:**

We thank the reviewers for their comments and will continue working on the subject.

**Withdrawal Confirmation:**

I have read and agree with the venue's withdrawal policy on behalf of myself and my co-authors.